# Genetic and karyotype divergence between parents affect clonality and sterility in hybrids

**Anatolie Marta[1]\*, Tomáš Tichopád[1], Oldřich Bartoš[2], Jiří Klíma[3], Mujahid Ali Shah[4], Vendula Šlechtová Bohlen[5], Joerg Bohlen[5], Karel Halačka[1], Lukáš Choleva[6], Matthias Stöck[7], Dmitrij Dedukh[1]\*, Karel Janko[1,6]\***

[1]Laboratory of Non-Mendelian Evolution, Institute of Animal Physiology and Genetics of the CAS, Libechov, Czech Republic; [2]Military Health Institute, Military Medical Agency, Prague, Czech Republic; [3]Laboratory of Cell Regeneration and Plasticity, Institute of Animal Physiology and Genetics of the CAS, Liběchov, Czech Republic; [4]Faculty of Fisheries and Protection of Waters, South Bohemian Research Center of Aquaculture and Biodiversity of Hydrocenoses, University of South Bohemia in Ceske Budejovice, Vodnany, Czech Republic; [5]Laboratory of Fish genetics, Institute of Animal Physiology and Genetics of the CAS, Liběchov, Czech Republic; [6]Department of Biology and Ecology, Faculty of Science, University of Ostrava, Ostrava, Czech Republic; [7]Leibniz-Institute of Freshwater Ecology and Inland Fisheries - IGB, Berlin, Germany

**\*For correspondence:**
anatolmarta@gmail.com (AM);
dmitrijdedukh@gmail.com (DD);
janko@iapg.cas.cz (KJ)

**Competing interest:** The authors declare that no competing interests exist.

**Abstract** Asexual reproduction can be triggered by interspecific hybridization, but its emergence is supposedly rare, relying on exceptional combinations of suitable genomes. To examine how genomic and karyotype divergence between parental lineages affect the incidence of asexual gametogenesis, we experimentally hybridized fishes (Cobitidae) across a broad phylogenetic spectrum, assessed by whole exome data. Gametogenic pathways generally followed a continuum from sexual reproduction in hybrids between closely related evolutionary lineages to sterile or inviable crosses between distant lineages. However, most crosses resulted in a combination of sterile males and asexually reproducing females. Their gametes usually experienced problems in chromosome pairing, but females also produced a certain proportion of oocytes with premeiotically duplicated genomes, enabling their development into clonal eggs. Interspecific hybridization may thus commonly affect cell cycles in a specific way, allowing the formation of unreduced oocytes. The emergence of asexual gametogenesis appears tightly linked to hybrid sterility and constitutes an inherent part of the extended speciation continuum.

## eLife assessment

This paper provides **important** insights into how asexual reproduction can arise in interspecific hybrids. The evidence supporting the conclusions is **compelling**, with rigorous molecular cytogenetic experiments showing the production of clonal gametes is common across hybrids between closely to moderately divergent sexual species. By highlighting the potential for asexuality to evolve in hybrids during a relatively wide window of species divergence, this work will be of broad interest to evolutionary biologists.

## Introduction

Speciation is a continuous process, often progressing with decreasing gene flow between diverging populations. According to the concept of the speciation continuum (*Shaw and Mullen, 2014*; *Stankowski and Ravinet, 2021*), hybridization between relatively closely related lineages typically results

**Figure 1.** Distribution of the spined loaches used in this study, and schematic representation of the reproduction in hybrids. (**a**) Phylogenetic tree based on exome-wide SNP data of the species used in the current crossing experiments. Red color indicates Bicanestrini group; green color indicates *Cobitis sensu stricto* group; yellow color indicates Adriatic group. (**b**) Distribution map of the spined loaches (*Cobitis*) included in this study. While *Cobitis elongatoides*, *C. taenia*, *C. tanaitica*, are known to act as parental species in hybridization events and emergence of clonal lineages, the other species (*C. strumicae*, *C. ohridana*, *C. taurica*, *C. pontica*) are known to be involved in secondary hybridization events. Abbreviations for all species in brackets. (**c**) Reproduction scheme of the clonal hybrids in the *C. taenia* complex, where E represents a haploid genome of *C. elongatoides* (orange), and T – of *C. taenia* (blue). Conventionally, hybridization between EE female and TT male lead to ET sterile males and clonal ET females that reproduce via gynogenesis. Gynogenetic females are pseudogamous, that is females produce diploid eggs via premeiotic genome endoreplication. Such eggs require sperm from a parental species (e.g. T sperm) to activate the embryonic development. The sperm genome is not incorporated in the hybrid's genome but is instead eliminated after the egg's activation, skipping karyogamy. All hybrid males are sterile because they do not produce spermatozoa or their extremely rare spermatozoa are aberrant and unable to fertilize eggs.

The online version of this article includes the following figure supplement(s) for figure 1:

**Figure supplement 1.** Phylogenetic relationships of the Cobitidae family based on the cytochrome b dataset (modified version of Perdices and co-authors (2016) *Perdices et al., 2016*).

**Figure supplement 2.** Karyotypes and karyograms of *C. ohridana* (**a, b**) and *C. tanaitica* (**c, d**).

**Figure supplement 3.** Plot of correlation between Castigla's AKD index (x-axis) and exome-wide genetic distance (SNP; y-axis).

in fertile and recombining progeny and may even lead to an evolutionary meltdown or homoploid hybrid speciation (*Comeault and Matute, 2018*). Increasing genetic divergence between hybridizing taxa may further promote the emergence of transgressive phenotypes (*Stelkens and Seehausen, 2009*) or hybrid vigour but typically lowers the mean fitness of hybrids, which consequently impacts the establishment of postzygotic reproductive isolating barriers (*Comeault and Matute, 2018*). In species with well-differentiated sex chromosomes, incompatibilities particularly affect the heterogametic sex (Haldane's rule) (*Haldane, 1922*; *Welch, 2004*; *Stöck et al., 2021*). Ultimately, speciation is completed when the accumulation of incompatible alleles in differentiating populations induces hybrid sterility or even inviability.

However, hybridization is also known to give rise to fertile, yet asexually reproducing hybrids in various animal and plant taxa. Asexual hybrids exhibit a broad spectrum of cytological mechanisms for the production of unreduced, often clonal, gametes, which range from entirely ameiotic processes (apomixis) to those involving more or less distorted meiosis (automixis; *Neaves and Baumann, 2011*; *Dedukh et al., 2022b*). A relatively common gametogenic alteration leading to asexuality is premeiotic genome endoreplication (*Figure 1*), found in a variety of hybrid fishes, amphibians, and reptiles (*Macgregor and Uzzell, 1964*; *Itono et al., 2006*; *Lutes et al., 2010*; *Dedukh et al., 2020*; *Dedukh et al., 2022a*). By auto-duplication, from each chromosome usually an identical copy is produced to pair with during the meiotic prophase. Premeiotic genome endoreplication thus not only ensures

clonal reproduction, but also allows hybrids to overcome problems in chromosome pairing that would otherwise lead to their sterility (*Dedukh et al., 2020*; *Janko et al., 2018*). Hybridization and asexuality thus represent important evolutionary phenomena (*Neaves and Baumann, 2011*).

To explain the apparent link between asexuality and hybridization, the 'balance hypothesis' has been proposed (*Moritz et al., 1989*), which assumes that incompatibilities, accumulated among parental genomes, may disturb gametogenesis in hybrids leading to the formation of unreduced gametes. Hybrid asexuality may thus emerge in a 'favourable evolutionary window' of genetic divergence between hybridizing species, which should be large enough to trigger particular gametogenic aberrations, but not too large to compromise the hybrid's fertility or even viability. The recently proposed concept of the 'extended speciation continuum' (*Stöck et al., 2021*) explicitly links the emergence of asexual reproduction to the speciation continuum. While forming species ultimately reach complete intrinsic reproductive isolation under increased divergence, fertile hybrids may be produced at earlier stages that exhibit aberrant gametogenesis, leading towards clonal reproduction. This not only facilitates the formation of allodiploid and allopolyploid asexual hybrid lineages, but asexuality per se also effectively restricts introgression between parental species due to non-recombinant production of gametes. Hybrid asexuality may thus be considered a particular type of Bateson-Dobzhansky-Muller (BDM)-incompatibilities, representing a form of intrinsic reproductive isolation during the speciation process (*Janko et al., 2018*).

While the evolutionary link between hybridization and asexuality has been empirically documented, it remains unclear how widespread such a link is and how commonly might asexual gametogenesis occur in hybrids. On the one hand, naturally occurring asexual hybrids exist in many major animal and plant lineages, but appear rare, suggesting that the genomic pre-conditions required for asexual reproduction may be rarely met during hybridization (*Stöck et al., 2010*). On the other hand, most of the knowledge about the evolution and biology of asexual hybrids comes from studies of naturally occurring lineages, while laboratory crossing experiments have been rarely performed to directly address the rate of clonal gametogenesis in hybrids. The questions how easily asexual reproduction is induced and why it is so often linked with a hybrid constitution thus remains unclear.

To estimate the incidence of clonal gametogenesis in a radiation of freshwater fish, we performed extensive crossing experiments of sexual species and investigated the gametogenic pathways in their hybrids, using European spined loaches of the family Cobitidae as a model. Cobitidae presents a speciose group of freshwater fish of Southeast Asian origin that spread over most of the Palearctic region since late Eocene (*Bohlen et al., 2019*). They colonized Europe in four distinct lineages (*Perdices et al., 2016*), which diverged from each other around 17–20 Mya (*Perdices et al., 2016*; *Majtánová et al., 2016*). The so-called *Adriatic* and *Bicanestrinia* lineages colonized Southern Europe and Near East regions, while Central and Eastern Europe have been colonized by the *Cobitis sensu stricto* lineage (*Figure 1a–b*). This lineage is composed of two sub-groups, the so-called *C. taenia* clade, involving several closely related species diverged during the last 1 Mya, and *C. elongatoides*, which diverged ~9 Mya from the *C. taenia* clade (*Janko et al., 2018*). Such independent colonization events and subsequent range shifts gave rise to natural hybrid zones particularly across Central and Eastern Europe (*Janko et al., 2012*). Diploid and polyploid hybrids have been reported within the family, including gynogenetic 'asexual' lineages. These lineages reproduce *via* gynogenesis, that is clonal eggs require sperm from sexual males to activate further development into genetically identical progeny. After fertilization, male pronucleus is usually eliminated from the eggs, without genetic contribution to the progeny (*Yamashita et al., 1990*; *Fyon et al., 2023*; *Figure 1a–c*). Such forms usually discard the sperms 'genome after fertilization and are currently found in hybrids of the Asian *Misgurnus anguillicaudatus* complex (*Morishima et al., 2008*), the *Cobitis hankugensis-longicorpa* hybrids in Korea (*Kim and Lee, 1995*), and in hybrids of the European *Cobitis sensu stricto* lineage (*Figure 1b*, *Choleva et al., 2012*). In these natural systems, only hybrid females are known to produce unreduced gametes, employing premeiotic genome endoreplication, while hybrid males are usually sterile due to aberrant pairing of chromosomes in meiosis (*Dedukh et al., 2020*; *Dedukh et al., 2021*).

In the present paper, we report meiotic and premeiotic gametogenic stages of experimental hybrids of eight loach species, representing three main European lineages. These species have been crossed in various combinations from closely related taxa to phylogenetically distant ones, which allowed us to test, how often unreduced hybrid gametes arise, whether their emergence is linked to genetic and karyotypic divergence between parental species, and how asexuality is related to hybrid sterility.

## Results

### Parental species and their karyotypes

The selected parental species represent three distinct phylogenetic lineages from Europe (*Figure 1a*, *Figure 1—figure supplement 1*; *Perdices et al., 2016*), possessing diverse karyotypes with diploid chromosome sets (2 n) between 48 and 50 chromosomes and variable numbers of meta-/submeta- and subtelo-/acrocentrics (*Figure 1—figure supplement 2*, *Supplementary file 1*). Karyotype dissimilarity among our focal species was measured as the autosomal karyotype index (AKD) which evaluates the differences in chromosome numbers and fundamental numbers of their arms (*Castiglia, 2014*). We found that AKD significantly increases with the genetic divergence among analyzed parental species, evaluated as the p-distances in SNPs in coding sequences (Mantel test, *r*=0.338, p=0.0261). However, the correlation was not always linear (*Figure 1—figure supplement 3*). This is because some phylogenetically distant species belonging to the Bicanestrinia and Adriatic phylogroups possess morphologically almost identical karyotypes (*Figure 1—figure supplement 2*, *Supplementary file 1*).

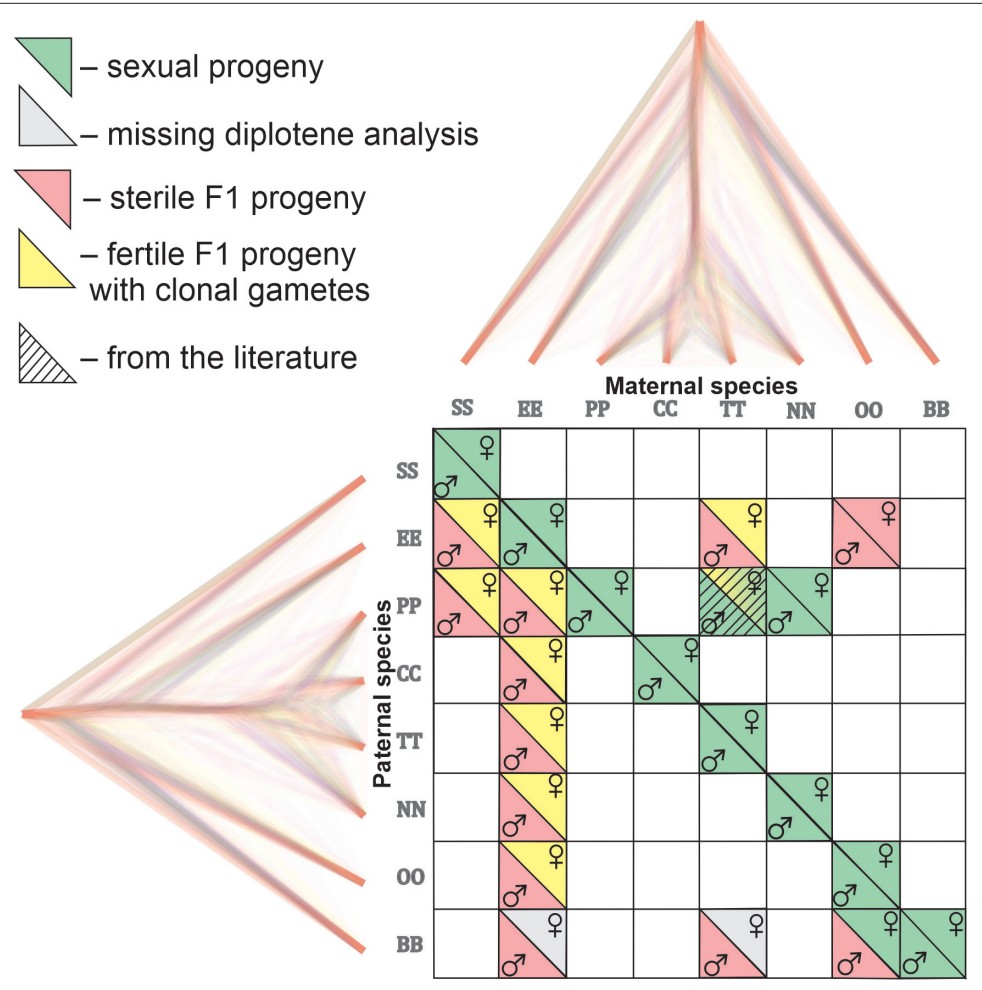

**Figure 2.** Matrix with successfully established and analyzed crosses between eight species of *Cobitis*. Each cell represents a particular cross between parental species including the information about maternal and paternal species (i.e. the direction of the cross). Phylogenomic tree plotted along margins indicates the relationships among crossed species based on exome-wide SNP data. Each color represents a particular reproductive output in F1 hybrids; green labeling indicates production of haploid gametes via normal meiosis; yellow color indicates the F1 progeny which produce unreduced gametes and present fully developed gonads; Red color denotes sterile progeny, predominantly referring to hybrid males. Grey labels potentially sterile hybrids for which we did not obtain diplotenic cells to fully confirm their reproductive output. The striped cell represents the F1 hybrid combination which was previously described *Janko et al., 2018* and involves the species used in this study.

## Artificial crosses and the presence of germ cells in all F1 hybrids

We crossed the eight focal species in 12 combinations, including some reciprocal crosses (*Figure 2*). To investigate the effect of increasing phylogenetic distance, we predominantly used EE males and females for crosses with most species belonging to the three sampled phylogroups, but species combinations without *C. elongatoides* were performed too. We obtained viable progeny of both sexes from all attempted species combinations, altogether resulting in 19 F1 hybrid families (*Figure 2*, *Supplementary file 2*). Here we present the abbreviation for every hybrid combination: *C. elongatoides ×C. taenia* (ET), *C. elongatoides ×C. tanaitica* (EN), *C. elongatoides ×C. taurica* (EC), *C. elongatoides ×C. pontica* (EP), *C. elongatoides ×C. strumicae* (ES), *C. elongatoides ×C. bilineata* (EB), *C. elongatoides ×C. ohridana* (EO), *C. taenia ×C. bilineata* (TB), *C. ohridana ×C. bilineata* (OB), *C. tanaitica ×C. pontica* (NP), *C. pontica x C. strumicae* (PS). We observed no significant deviations from equal sex ratios among the F1 (Binomial test, all P values >0.25 after corrections for multiple testing; *Supplementary file 2*) and - about 6 months after hatching -randomly selected 3–5 juveniles per family to examine the gametogenesis in F1 hybrids.

Whole-mount immunofluorescence staining of gonads with antibodies against *Vasa* protein showed that the analysed F1 hybrids contained germ cells organized in clusters, which were similar to germ cell clusters of sexual fish, used as a control group (*Figure 3—figure supplement 1a, c, e*). The immunofluorescence staining of SYCP3 protein was performed either on whole-mount staining of gonadal tissue or on pachytenic spreads, and revealed the formation of lateral components of synaptonemal complexes (SCs) in most controls and F1 hybrids, suggesting the presence of meiocytes (*Figure 3*, *Figure 4*, *Figure 3—figure supplement 1b, d, f*, *Figure 3—figure supplement 2*, *Figure 4—figure supplement 1*, *Figure 4—figure supplement 2*). The only exceptions were PS and TB hybrid juvenile males, whose gonads contained clusters of germ cells without meiocytes. The absence of meiocytes was later verified also in both, adult PS and TB males (*Figure 3—figure supplement 1a, b*).

## Analysis of meiocytes

We then investigated chromosome pairing during the pachytene stage of the meiotic prophase by immunostaining of the SYCP3 and SYCP1 components of the synaptonemal complex, and we analysed lampbrush chromosomes of the diplotene meiotic stage in females. We analysed 24 parental individuals, including males and females (EE, TT, NN, PP, OO) and 79 individuals among all F1 families (*Supplementary file 2*, *Supplementary file 3*).

### Sexual species produce reduced gametes with properly formed bivalents

Depending on their chromosome numbers, parental species always contained 24–25 synaptonemal complexes (*Figure 3a*, *Figure 4a*) in 852 analysed cells (*Supplementary file 3*). Flow cytometry of testicular cell suspensions from parental species revealed the presence of haploid and diploid cells with a major peak corresponding to haploid cells (*Figure 3—figure supplement 3a*). Histological analysis showed many sperm cells organized in cysts in all sexual males investigated, containing properly developed Sertoli and Leydig cells (*Figure 3—figure supplement 4a*), suggesting normal development of testes.

### Sterility of hybrid males is caused by both, the aberrant pairing in the meiotic prophase and the inability to proceed to meiosis

Flow cytometry revealed that the analysed hybrid males lacked the haploid population of cells, characteristic for spermatids and spermatozoa of sexual males. Hybrid males had only diploid cell populations (*Figure 3—figure supplement 3b–h*). Interestingly, TB males had a small population of diploid cells corresponding to pachytene stage. These males had morphologically underdeveloped gonads, indicating that here the sterility is probably caused by the inability of gonocytes to proceed to meiosis (*Figure 3—figure supplement 1a, b*; *Figure 3—figure supplement 3h*). Histological examinations demonstrated that analysed hybrid males had defective testes with an asynchronous development of germinal cells in cysts (*Figure 3—figure supplement 4b–g*). Similar results were reported in naturally occurring ET hybrid males *Dedukh et al., 2020*.

Furthermore, hybrid males from most experimental crosses showed aberrant chromosomal pairing during the pachytene in 244 analysed cells (*Supplementary file 3*). Abnormal cells had variable

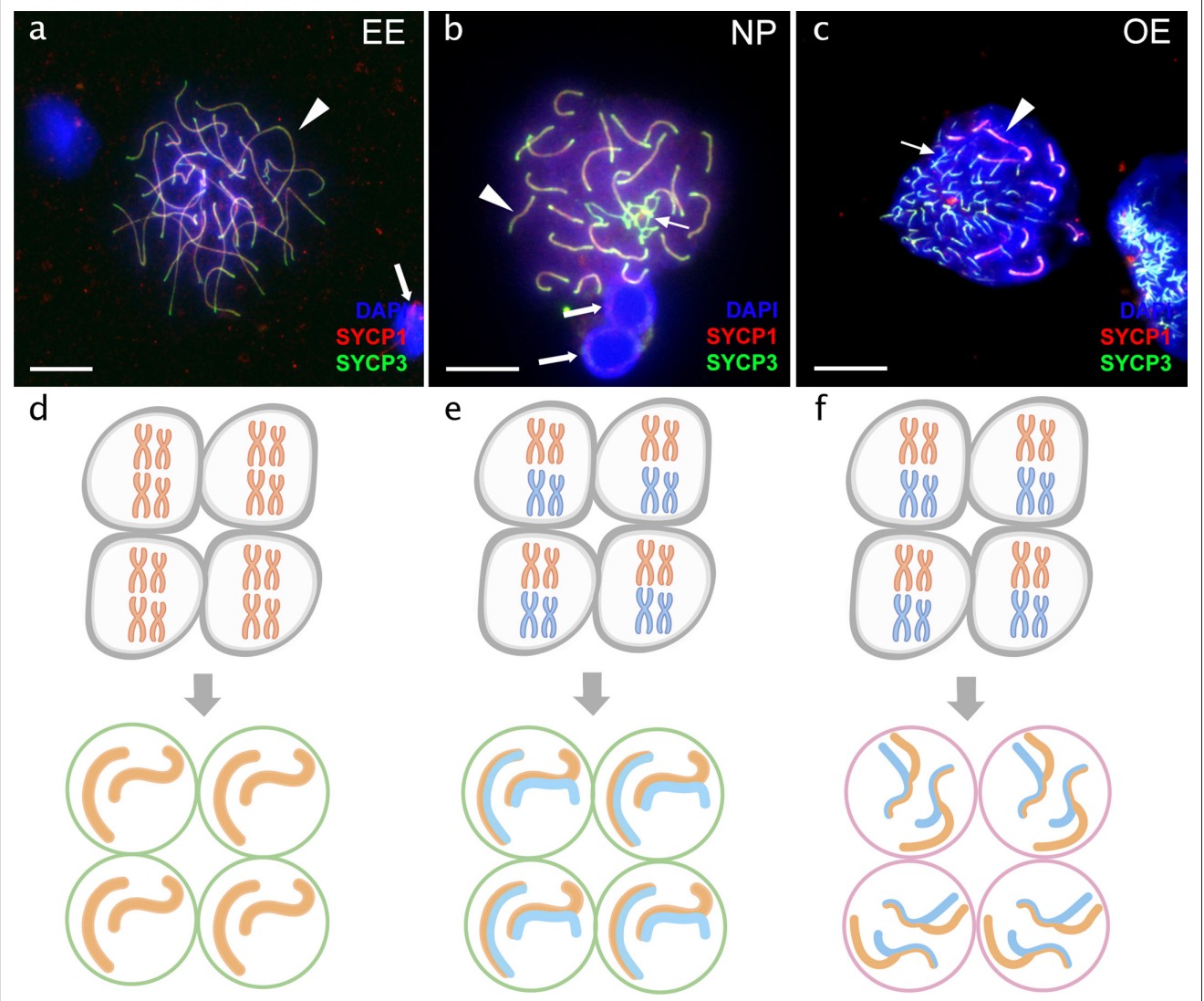

**Figure 3.** Pachytene spermatocytes in males. Comparison of pachytene spermatocytes between male of sexual diploid species (**a**), hybrids' genotypes (**b, c**), and corresponding gametogenic pathways (**d-f**). The spread of pachytene spermatocytes of *C. elongatoides* with 25 completely paired bivalents (**a**), and diploid NP hybrid with almost paired chromosomes (**b**) and OE hybrid with aberrant pairing, including bivalents and univalents (**c**). Thick arrows indicate bivalents; thin arrows indicate univalents. Scale bar = 10 μm. Schematic representation of the gametogenic pathway including presumptive karyotype composition in gonocytes and pachytene cells in males of sexual species (**d**) and NP hybrids (**e**) that are able to complete pairing and form gametes meiotically, and OE hybrids (**f**) which exhibit abnormal pairing leading to sterility.

The online version of this article includes the following figure supplement(s) for figure 3:

**Figure supplement 1.** Visualization of gonial and meiotic cells in male gonads.

**Figure supplement 2.** Pachytene spermatocytes of hybrid males.

**Figure supplement 3.** Relative DNA content of cell nuclei from testes measured by flow cytometry.

**Figure supplement 4.** Histological examination of testes.

numbers of univalents, bivalents and multivalents (*Figure 3c*; *Figure 3—figure supplement 2a–g*). The only exception to aforementioned observations were the NP hybrids involving closely related parental species (*C. tanaitica* and *C. pontica*) where we detected fully-paired chromosomes during the pachytene, in 36 cells (*Figure 3b*). Chromosomal spreads of NP males also clearly showed the presence of formed spermatid nuclei, similar to sexual males (*Figure 3b*). Additionally, we counted SCs, based on the presence of crossing-over foci (*Figure 4—figure supplement 2*; *Dedukh et al., 2020*; *Sun et al., 2006*).

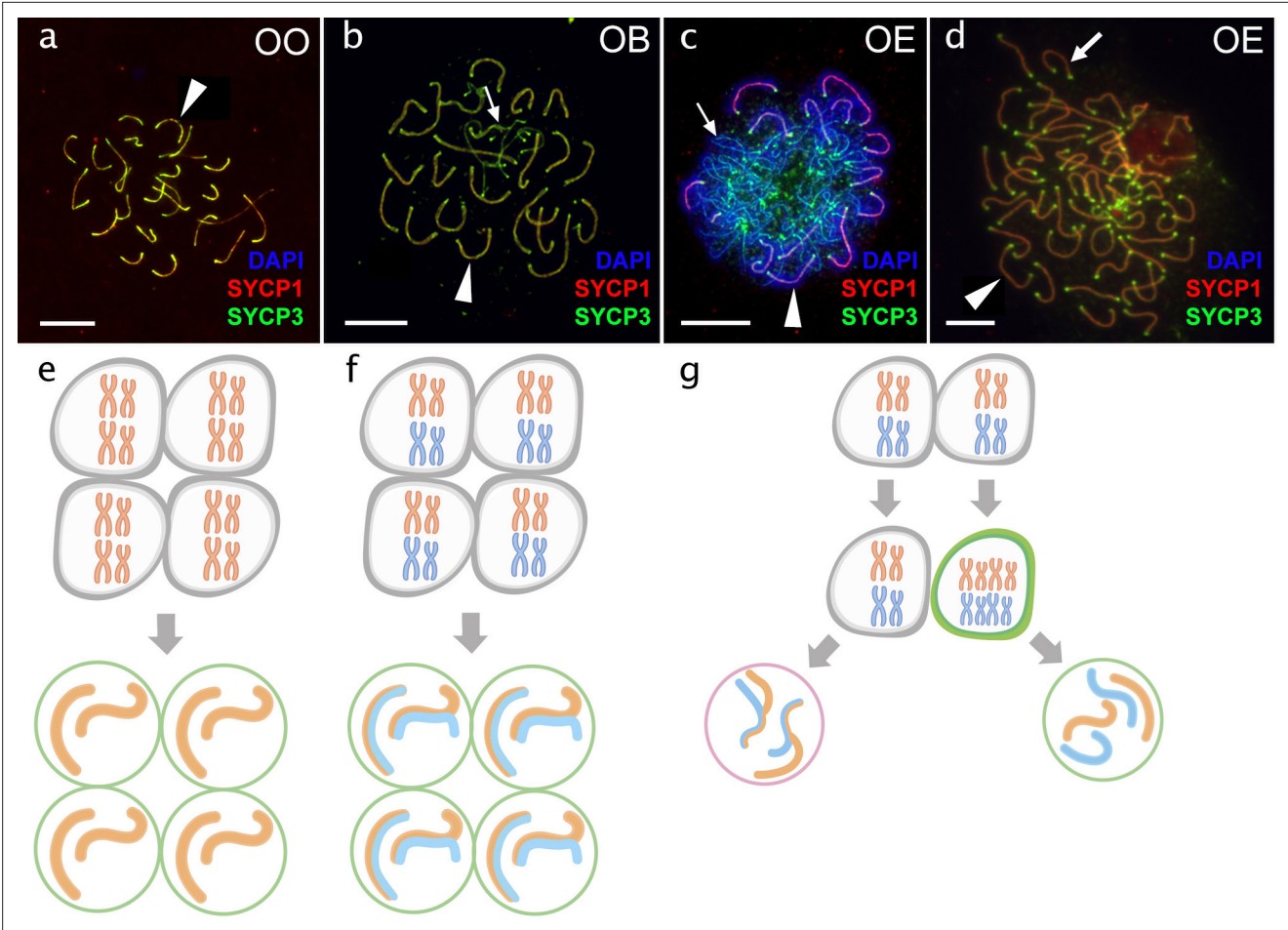

**Figure 4.** Pachytene oocytes in females. Comparison of pachytene oocytes between female of sexual diploid species (**a**) and hybrids genotypes (**b, c, d**) and corresponding gametogenic pathways (**e–h**). The SC spreads from pachytene oocyte of *C. ohridana* with fully paired 25 bivalents (**a**), diploid OB hybrid with almost paired chromosomes (**b**) and OE hybrid female which have pachytene cells with aberrant pairing (**c**) and cells with fully paired chromosomes emerged after premeiotic genome endoreplication (**d**). Thick arrows indicate bivalents; thin arrows indicate univalents. Scale bar = 10 µm. Schematic representation of gametogenic pathway including presumptive karyotype composition in gonocytes and pachytene cells. Females of sexual species (**e**) and OB hybrid (**f**) which are able to fully or partially complete pairing of chromosomes and form gametes meiotically; OE hybrids (**g**) exhibit two populations of pachytene oocytes: oocytes with unduplicated genomes and oocytes with duplicated genome. Oocytes with unduplicated genome (g, left) have abnormal pairing leading to the inability of proceed beyond pachytene and thus sterility. Oocytes with duplicated genomes (g, right) have normal pairing and thus leading to the formation of unreduced gametes.

The online version of this article includes the following figure supplement(s) for figure 4:

**Figure supplement 1.** Pachytene oocytes with nonduplicated genomes.

**Figure supplement 2.** Pachytene spreads of sexual species and hybrids.

**Figure supplement 3.** Chromosomal spreads from diplotene oocytes of hybrid females.

### Production of unreduced gametes by premeiotic genome endoreplication is the prevalent gametogenic pathway among hybrid females

Analysis of diplotenic oocytes of most hybrid females revealed no univalents or mispaired chromosomes, but instead showed the formation of 49 or 50 properly formed bivalents (*Figure 4—figure supplement 3a–e*). Similar to previous data on natural clones *Itono et al., 2006*; *Dedukh et al., 2021*, we therefore conclude that diplotene oocytes in the majority of hybrid females contain only cells with duplicated genomes, suggesting their ability to complete clonal gametogenesis (*Figure 4*; *Figure 4—figure supplement 3a–e*). The only exceptions were observed in OB and NP hybrids, where diplotenic oocytes contained only 25 fully formed bivalents, suggesting normal meiosis with the pairing likely between orthologous chromosomes (*Figure 4—figure supplement 3f*).

However, the analysis of 665 pachytenic oocytes revealed different patterns (*Supplementary file 3*). OB and NP females possessed 25 bivalents with delayed pairing of two chromosome pairs (*Figure 4—figure supplement 1d*), generally matching their diplotene configurations. However, most F1-females contained pachytenic cells (n=535) with abnormally-paired chromosomes and various numbers of univalents, bivalents, and multivalents (*Figure 4c*, *Figure 4—figure supplement 1a–c and e–h*). This suggests that most pachytenic oocytes in hybrids did not undergo genome endoreplication. Pachytenic oocytes with duplicated genomes and thus normal chromosomal pairing were only observed in females from three F1-families (EC, EO, and ET hybrids). These showed sets of 50 or 49 bivalents with properly loaded lateral and central synaptonemal components and MLH1 loci on each bivalent (*Figure 4d*).

Overall, the incidence of genome endoreplication was very low among pachytenic cells of any F1 hybrid female, occurring on average in 0.7% of cells only (*Supplementary file 3*) and there were no significant differences between any combinations of parental species (ANOVA Chi square test of generalized linear mode (GLM); DF = 9, dev.=7.5, P-value = 0.59). We also did not observe any significant effects of maternal species in those crosses, where both directions were performed, i.e. there were no obvious differences between ET and TE as well as between OE and EO cross types.

We therefore treated all experimental progenies as a single group and tested whether they differed from previously-analysed natural diploid ET-hybrids, whose incidence of duplicated gonocytes was higher, on average ~6%; *Dedukh et al., 2021*. Here, the difference was significant (ANOVA Chi square test of GLM; DF = 1; dev = 22.07; p-value <$10^{-5}$), suggesting that natural clones produce a higher proportion of oocytes with endoreplicated genomes than experimental F1 hybrids.

The contrasting incidence of oocytes with endoreplicated genomes between pachytene (approx. 6%) and diplotene stages (100%) is consistent with previous results *Dedukh et al., 2021* suggesting that premeiotic genome endoreplication is a rare event in all types of crosses. This allows cells to form normal bivalents and to enter the diplotene, while the majority of cells remain non-duplicated and cannot proceed beyond the pachytene checkpoint due to mispairing among orthologous chromosomes.

Importantly, we noted that incidence of duplicated gonocytes was significantly higher among F1 hybrids (treated as a single group), than in females of any tested parental species, where in fact no duplications have been observed among 792 analyzed oocytes (ANOVA Chi square test of GLM; DF = 1; dev = 7.506; p-value = 0.00615).

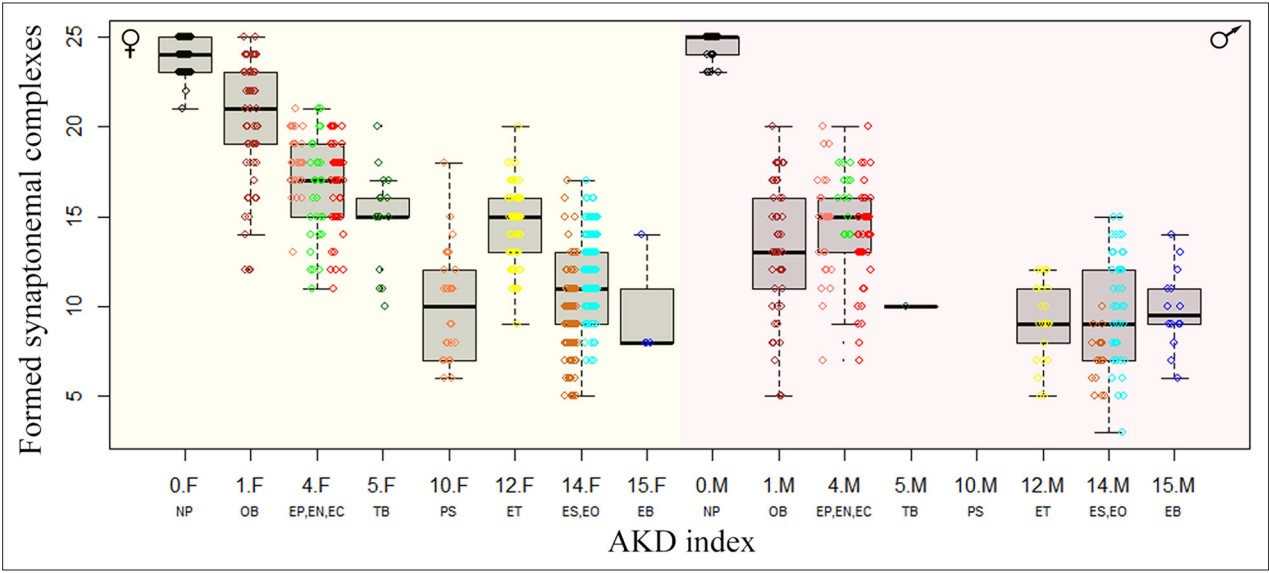

**Figure 5.** Effect of karyotype dissimilarity on numbers of bivalents in hybrids. Boxplots showing the number of synaptonemal complexes per cell (y-axis) of the F1 hybrids ranked along x-axis according to the morphological divergence of karyotypes between parental species expressed by AKD index (*Castiglia, 2014*). Numbers indicate the AKD index values, subscripts F and M indicate females and males, respectively and experimental crosses are indicated by respective letter combinations.

## Divergence of parental karyotypes affects the formation of bivalents between orthologous chromosomes in hybrids but has different effects in males and females

We than focused exclusively on those pachytenic cells with unduplicated genomes and found that F1-hybrids largely vary in numbers of paired orthologous chromosomes, ranging from three detectable synaptonemal complexes in EO males to 25 fully-formed bivalents in OB females. To explain such a variability, we fitted the distribution of numbers of synaptonemal complexes by variables assuming (a) sex of the hybrid individual, (b) chromosomal divergence between its parental species measured through the AKD index and (c) their genetic divergence measured in exome-wide SNP p-distance. All three variables significantly contributed to the data distribution with numbers of bivalents in pachytenic cells being negatively correlated with morphological dissimilarity of parental karyotypes (GLMM z value = –5.86; p<10$^{-8}$) and with their genetic distance (GLMM z value = –3.73; p<10$^{-3}$). Males also tended to have fewer bivalents than females (z=–4.54; p<10$^{-5}$). The model also suggested that both slopes are steeper in males than in females, however, the interaction of AKD or SNP divergence with hybrids' sex was not significant (*Figure 5*). This suggests that the formation of bivalents in hybrids is negatively affected by chromosomal dissimilarity as well as overall genetic divergence between parental species, and that male offspring generally form less bivalents than female progeny.

## Comparison with other crossing experiments within Cobitidae

Finally, to put our results in context with previously published data, we plotted the reproductive outputs of all known loach hybrids in relation to genetic divergence and chromosomal differences between their parental species. To express the amount of chromosomal dissimilarity among parental species, we used the AKD index calculated from published and here newly prepared karyotypes of all

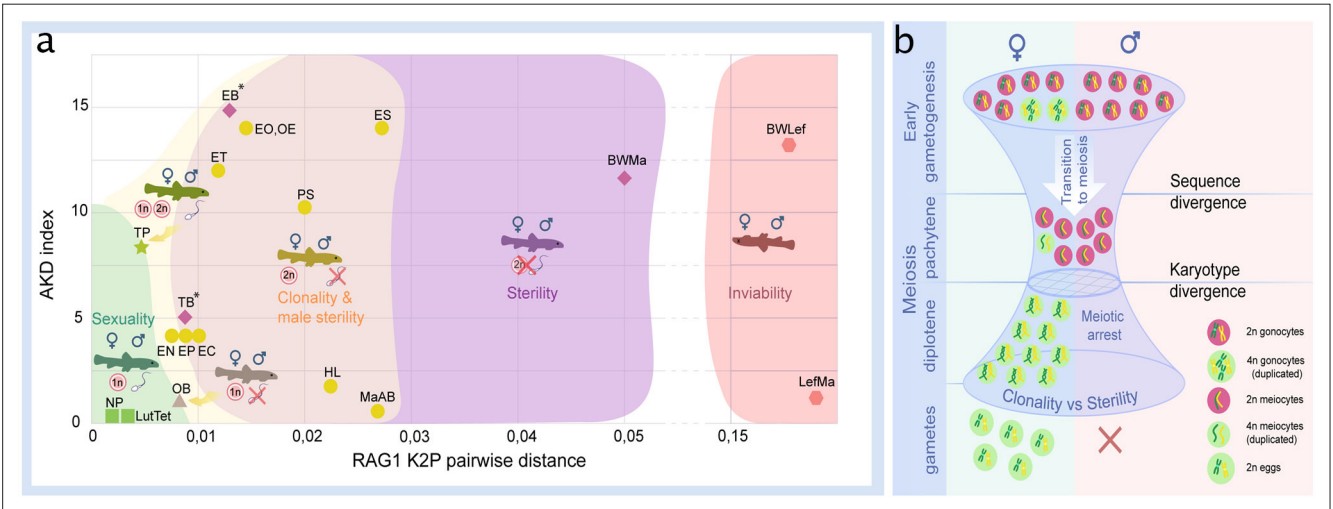

**Figure 6.** Reproductive outcomes of hybrids from crosses between species from the subfamily Cobitinae. (**a**) Plot demonstrating the relationship between reproductive outcome and viability of F1 hybrid loaches and the genetic and karyotype divergences among parental species. Data on F1 hybrids have been obtained in this study as well as from the literature. Karyotype divergence is marked as AKD index and genetic distance is estimated from published RAG1 sequences (K2P distance). Green color shows the ranges of hybrids with sexual reproduction; yellow color shows the ranges of F1 hybrids, which produce unreduced gametes; purple color indicates the ranges of sterility in both sexes; red color shows the ranges of inviable hybrids. To rank reproductive outcomes for every type of cross we labeled them with additional symbols: ■ – both sexes producing haploid gametes with fully paired chromosomes; ★–both sexes are fertile producing haploid gametes, some females produce unreduced gametes; ▲ – females produce haploid eggs, males are sterile; ● – females produce clonal eggs, males are sterile; ♦ – both sexes are sterile; ● – both sexes are inviable. Asterisk (*) indicates cross combinations for which diplotene analysis is missing. Each abbreviation represent a subgenome of the parental species: *Cobitis elongatoides* = EE; *C. taenia* = TT; *C. tanaitica* = NN; *C. taurica* = CC; *C. pontica* = PP; *C. strumicae* = SS; *C. bilineata* = BB; *C. ohridana* = OO; *C. hankugensis* = HH; *C. biwae* = BW; *C. lutheri* = Lut; *C. tetralineata* = Tet; *Iksookimia longicorpa* = LL; *M. anguillicaudatus* clade A=MaA; *M. anguillicaudatus* clade B=MaB; *Lefua echigonia* = Lef. Hypothetical scheme of the selective processes necessary for the emergence of clonality in F1 hybrids (**b**). First step includes the accumulation of mutations in gametogenic regulatory genes which is required to induce premeiotic genome endoreplication in F1 hybrid females. Second step includes the chromosomal divergence which leads to the aberrant chromosomal pairing followed by the cell arrest of meiocytes with non-duplicated genomes. Premeiotic genome endoreplication is sex specific, while hybrid males are sterile.

spined loaches known to hybridize. To express the amount genetic divergence among these species, we used their sequence distances in the nuclear RAG1 gene, as a proxy, because exome-based sequence data are not available for some of the previously examined crosses. Data on RAG1- and karyotype divergences were then correlated with reproductive outputs in their hybrids and presented on *Figure 6a*.

Inviable or fully sterile hybrids were generally found among hybrids between the genetically most distant species, while sexual and/or clonal reproduction were observed in hybrids between more closely related species. However, the window of genetic divergences allowing for sexual as well as asexual reproduction was relatively wide (*Figure 6a*). Interestingly, we noticed that asexuality tends to be observed among female hybrids between species with substantially diverged karyotypes, while sexual reproduction generally required more similar karyotypes.

## Discussion

Using freshwater fishes, loaches (Cobitidae), as a model, the present study reveals a wide range of reproductive outcomes in F1 hybrids in relation to genetic and karyotype divergences between the parental species (*Figure 6*). On one extreme, hybrid inviability has been reported in crosses between distant species (*Suzuki, 1957*). On the other extreme, the production of haploid gametes was found in crosses of closely related species, such as between C. lutheri and C. tetralineata (*Kwan et al., 2019*), and between C. pontica and C. taenia *Janko et al., 2018* or C. tanaitica species, whose divergence has been dated to ~1 Mya *Janko et al., 2018*, but also in OB female hybrids in a relatively distant species pair (*Figure 2*, *Figure 6*, *Figure 4—figure supplement 1d*, *Figure 4—figure supplement 2d*). However, the most common types of reproductive outputs were hybrid sterility and production of unreduced gametes.

In case of hybrids between relatively far-distant species pairs, like the intergeneric *Cobitis-Misgurnus* cross, or between Adriatic and *Cobitis sensu stricto* lineages (i.e. EB and TB hybrids), sterility occurred in hybrids of both sexes (*Figure 6*). However, the most common outcome was asymmetric, with sterility affecting hybrid males, while F1 females produced unreduced oocytes. This pattern occurred in the majority of crosses, including species pairs, like SE, SP or EO, with divergences reaching ~17 or even >20 Mya back (*Majtánová et al., 2016*; *Figure 2*, *Figure 6*) and it has also been reported from several natural hybrid complexes (*Itono et al., 2006*; *Dedukh et al., 2020*; *Dedukh et al., 2021*). An intermediate situation was observed in TP-hybrids between a closely related species that diverged ~1 Mya (*Janko et al., 2018*), where males produced reduced gametes and females produced both, reduced and unreduced ones.

It is generally assumed that the emergence of asexuality is a rare phenomenon that may require complex changes in gametogenic pathways and reproductive modes (*Stöck et al., 2010*). This opinion has been corroborated by the 'twiggy' distribution of 'asexual' organisms on the tree of life (*Schwander and Crespi, 2009*) and by many failures to re-generate naturally occurring asexual hybrids by experimental crosses of their parental species (e.g. in apomictic *Poecilia* fish and parthenogenetic *Darevskia* lizards *Stöck et al., 2010*; *Murphy et al., 2000*). However, we observed the production of unreduced gametes in 7 out of 11 crossed species pairs of loaches and in 10 out of 18 species pairs when including literature reports and natural hybrids (*Figure 2*, *Figure 6*). Such a widespread emergence of asexual gametogenesis therefore appears not to depend on the presence of a genome from any particular parental species, but to arise relatively commonly among independent species pairs across the entire group of spined loaches. Asexual gametogenesis has also been observed in experimental F1 progeny of several other vertebrate species (*Iwai et al., 2011*). This suggests that the formation of unreduced gametes, that is the necessary prerequisite for asexual reproduction, may be a much more common outcome of interspecific hybridization than it would appear from the rarity of asexual hybrid taxa that are successfully established in nature.

Some studies also suggested that the emergence of asexuality requires the evolution of specific genes within hybrid lineages after their origin (*Alberici da Barbiano et al., 2013*). However, our finding that asexual gametogenesis originates relatively commonly already in the F1 generation somewhat refines this hypothesis and suggests that it is a direct result of genome merging by inter-specific hybridization. Importantly, all F1-females capable of clonal gametogenesis employed an identical cytogenetic mechanism, i.e. premeiotic genome endoreplication, as also observed in many natural asexual vertebrates (*Betto-Colliard et al., 2018*; *Itono et al., 2006*; *Dedukh et al., 2020*;

*Dedukh et al., 2022a*; *Dedukh et al., 2021*; *Iwai et al., 2011*; *Bi and Bogart, 2010*; *Stöck et al., 2012*) suggesting that gametogenic perturbations induced by hybridization are often canalised into a similar developmental pathway. The reasons for such common patterns are not clear. The regulation of cell cycle is particularly sensitive to stress and perturbations (*Rotelli et al., 2019*) and interspecific hybridization may act as such stimulus (*Moritz et al., 1989*). While asexual reproduction may employ various pathways (*Neaves and Baumann, 2011*), it is possible that aberrations of cell cycle leading to endoreplication are more common since they require less accumulated incompatibilities than other routes to the production of unreduced gametes. Alternatively, the occurrence of endoreplication may reflect some predispositions, like the spontaneous production of unreduced gametes in some sexual parental lineages, which would then ensure clonal reproduction in their hybrids. However, our data argue against this alternative, since we analysed hundreds of gonocytes of sexual females without a single incidence of genome endoreplication (*Dedukh et al., 2020*; *Dedukh et al., 2021*). Moreover, during years of crossing experiments and population sampling, we genotyped hundreds of *C. elongatoides* and *C. taenia* and their progenies and never found an autotriploid specimen (*Janko et al., 2012*; *Tichopád et al., 2022*), which would be expected if fertilisation of unreduced gametes existed in these species (*Flajšhans et al., 2007*).

Our in-depth analysis of hybrid gametogenesis in *Cobitis* loaches further provided compelling evidence for a tight link between hybrid incompatibility, asexuality and sterility. Namely, the majority of meiocytes produced by both the male and female hybrids failed to form a full set of bivalents between orthologous chromosomes and were arrested at meiotic checkpoints (*Figure 3*). However, unlike their sterile brothers, hybrid F1 females also usually contained a minor proportion of oogonia that underwent premeiotic genome endoreplication. In most our F1 hybrid females, only these rare oogonia were able to complete meiosis and form diploid gametes. Interestingly, similar observations were reported from several natural and experimental hybrid vertebrates, like ET and EN hybrid loaches, geckos and whiptail lizards (*Aspidoscelis*) Medaka ricefish (*Oryzias*) (*Dedukh et al., 2022a*; *Dedukh et al., 2021*; *Iwai et al., 2011*; *Newton et al., 2016*). This suggests that even successfully established hybrid asexuals usually contain only a minor proportion of oogonia, which actually undergo clonal gametogenetic pathway.

Such congruent patterns across phylogenetically distant taxa thus conform to the concept of an 'extended speciation continuum'. Similar to the classical concept of the 'speciation continuum', it assumes that speciation generally proceeds from early stages characterized by the production of sexual hybrid progeny to irreversibly diverged species, whose hybrids are either sterile or even inviable. At intermediate stages, however, hybrids between diverging evolutionary lineages may not only have a certain incidence of sterile gonocytes, but also tend to produce unreduced gametes (*Figure 6*). Genome endoreplication thus appears as an effective way to overcome sterility in hybrid females, even at high levels of divergence. Asexuality may thus often represent the only reproductive pathway of hybrids between a given pair of species and, together with hybrid sterility, it may simultaneously contribute to reproductive isolation of parental species due to clonal genome propagation.

We observed that the fertility of female hybrids remains conserved due to sex-specific emergence of clonality, while males of the same cross are usually sterile. This may potentially indicate some analogy to other classical sex-related asymmetries in hybrids, like incompatibilities between sex chromosomes (Haldane's rule) or other uniparentally inherited factors (Darwin's corollary *Brandvain et al., 2014*). However, the sex determination systems are not yet known in most of the investigated species (*Stöck et al., 2021*) and in any case, the reasons for such an asymmetry are likely more complex. For instance, meiocytes in hybrid males contained significantly lower numbers of properly-formed synaptonemal complexes than non-duplicated meiocytes in females (*Figure 5*) and were more severely affected by decreasing similarity of parental karyotypes (see Results, 2.3.5.). This indicates some more fundamental differences in pairing mechanisms and affinity among orthologous chromosomes fundamentally between spermatogenesis and oogenesis, e.g. (*Blokhina et al., 2019*). In fact, we recently found that spermatogonial cells taken from a sterile ET hybrid male may resume the ability of endoreplication when transplanted into female gonads (*Tichopád et al., 2022*). This suggests that initiation of premeiotic endoreplication may rely on female-specific gonadal environment rather than on genetic sex determination.

The genetic divergence among parents may affect early gametogenic stages in hybrids through various mechanisms causing for example depletion of primordial germ cells (*Yoshikawa et al., 2018*),

improper pairing of diverged chromosomes (*Ferree and Prasad, 2012*), asynapsis between sex chromosomes (*Bhattacharyya et al., 2013*), or deleterious epistatic interactions between orthologous genes and their binding motifs on chromosomes (*Balcova et al., 2016*). In loaches, we documented at least two such mechanisms underlying hybrid sterility. TB and PS hybrid males had almost no spermatocytes, suggesting their gonocytes probably failed the transition to meiosis. The other type of sterility conforms to a chromosomal speciation model (*Faria and Navarro, 2010*; *Potter et al., 2017*), prevailing in most F1 hybrid males, in which spermatocytes developed but possessed aberrantly paired chromosomes, preventing further progression to spermatids. Analogous patterns were observed in non-duplicated oocytes of most hybrid females, suggesting that problems in bivalent-formation have a detrimental impact on the meiosis in both male and female loach hybrids.

Karyotype evolution among parental species indeed plays a crucial role in hybrids' reproductive capabilities (*Potter et al., 2017*). Our data suggest that it has a crucial, yet indirect, impact on asexual reproduction. Specifically, hybrids between species with diverged karyotypes had less bivalents in non-duplicated meiocytes than those combining similar karyotypes. Sometimes, even hybrids between phylogenetically distant, yet karyotypically similar species could produce full sets of bivalents (e.g. OB hybrids; *Figure 5*). On the other hand, the numbers of endoreplicated oocytes were low in any hybrid. Their proportion was relatively higher in natural clones than in experimental F1 *Cobitis*, suggesting that such a trait may recover during the evolutionary establishment of a natural asexual lineage (*Dedukh et al., 2021*) and/or that selection among 'freshly' emerged clones would favour those with the highest proportion of endoreplicated oocytes. Nevertheless, even the most widespread clones possessed only ~6% of such oogonia, suggesting that karyotype similarity allows the completion of meiosis and hybrids would mostly reproduce sexually by reduced gametes, even if they had the capacity of producing unreduced gametes, too.

While hybridization is assumed to present an important trigger of asexuality in vertebrates and other organisms (*Neaves and Baumann, 2011*), the origins of clonal lineages are thought to be rare, and often can be traced back to a single or few hybridization events in a given species pair (*Stöck et al., 2010*; *Choleva et al., 2012*). Multiple independent origins of clonal gametogenesis in spined loaches (*Janko et al., 2012*; *Choleva et al., 2012*, this study) in part contradicts this view since interspecific hybridization has been found to result in clonal gametogenesis relatively commonly up to a substantial divergence, where fertile or viable hybrids are no longer appearing (*Figure 6a*). However, we emphasize that successful establishment of hybrid clones under natural conditions is likely restricted by several additional levels of selective pressure and thus filtering (*Figure 6b*). On the one hand, the 'balance hypothesis' (*Moritz et al., 1989*) and its expansions and fine-tuned versions (*Janko et al., 2018*; *Stöck et al., 2010*), suggest that the production of unreduced gametes probably requires some level of genetic divergence among the parental species to alter the hybrid cell cycle towards genome endoreplication. On the other hand, it appears that most meiocytes of asexual hybrids anyway do not pass through endoreplication (*Dedukh et al., 2022a*; *Dedukh et al., 2021*; *Iwai et al., 2011*; *Newton et al., 2016*). Consequently, if the parental karyotypes remain largely similar, then orthologous chromosomes may successfully pair even in such nonduplicated meiocytes, potentially, allowing them to accomplish normal meiosis and form reduced gametes. This would decrease the proportion of unreduced gametes produced by such hybrids and negatively impact on their likelihood to establish a stable asexual lineage.

We therefore propose, that for a cross to be successful in generating an asexual hybrid, it not only has to produce a sufficient number of unreduced gametes, but it also has to prevent the formation of reduced gametes from non-duplicated meiocytes. One way to achieve this may be if hybridization involves species with sufficiently diverged karyotypes, ensuring improper pairing of orthologous chromosomes in non-duplicated gametes. Increased divergence of parental karyotypes may therefore have a positive impact on the production of stable clonal hybrids and, for instance, the fixation of certain chromosomal rearrangements among diverging populations may be accelerated by certain geographical aspects of speciation, such as parapatry (*Faria and Navarro, 2010*; *Potter et al., 2017*). It would thus be attractive to test if the incidence of asexual hybrids correlates with increased rates of karyotype evolution among parental species.

The rarity of naturally occurring clonal organisms probably results from the simultaneous impact of the abovementioned selective processes. However, our data indicate that the production of unreduced gametes, which is the very basic *conditio sine qua non* for the evolution of asexuality, may be

surprisingly 'easy' to trigger, especially in hybrids of distantly related lineages. Of course, we may not rule out that the frequent emergence of premeiotic endoreplication in Cobitidae reflects some specific predispositions in their gametogenic machineries in comparison to other taxa, where asexuality appears rarer. Thus, our study demonstrates that conclusions about the evolution of asexuality should not only be drawn from natural asexuals, but should be based also on experimental crosses.

## Materials and methods
### Selection of parental specimens and genotyping
All *Cobitis* specimens were obtained from the rearing facility stocks of the Laboratory of Fish Genetics, IAPG CAS CZ that had been collected during recent projects across Central Europe in accordance with environmental protection legislation. The Valid Animal Use Protocol was in force during the study at the IAPG (No. CZ 02386). All institutional and national guidelines were covered by the 'Valid Animal Use Protocol' No. CZ 02386 of the Laboratory of Fish genetics, IAPG CAS. For the breeding experiments, we selected 38 individuals of 8 species. Taxonomic identification and genotyping of examined individuals were based on previously determined and routinely applied molecular markers involving two Sanger-sequenced nuclear markers (the intron in S7 gene and RAG1 gene), and one mitochondrial gene, cytochrome b (CytB), which were compared to previously published data to confirm their taxonomical identification using routine protocols described in *Janko et al., 2012*. Experimental design of crosses depended on available fish in our aquariums. So far, the most abundant, in males and females, were EE, which was bred with all other species. In the case of breeding of TT with PP, BB, crosses were unique or had only two replicates because of the limited number of available fish.

To obtain phylogenomic data and evaluate SNP, we applied the exome-capture approach to all experimentally crossed lineages (*Janko et al., 2018*). Briefly, isolated genomic DNA was sheared with Bioruptor, tagged by indices, hybridized to custom-designed exome-capture probes (*Janko et al., 2018*) sequenced with 2*75 mode on Illumina NextSeq device. Raw data were processed (adapter and quality trimming using *bbduk.sh* script from BBmap package *Bushnell, 2014*; *ktrim = r k=23 mink = 11 hdist = 1 tpe tbo qtrim = rl trimq = 10 maq = 10 minlen = 36*). Processed reads were aligned to *C. taenia* reference transcriptome that was published and cleaned from potentially paralogous contigs by *Janko et al., 2018*. Mapping was performed with *bwa mem* algorithm *Li, 2013* in default settings and SNP calling with GATK4 pipeline *Auwera and O'Connor, 2020* (indels were excluded).

### Phylogenetic inference and estimation of pairwise genomic distances
Mitochondrial loci were eliminated from all downstream analyses of SNP, which focussed only on nuclear exon sequences. The VCF dataset was used to calculate pairwise p-distances in SNPs between all individuals using VCF2Dis v1.47 software (*Subramanian et al., 2019*).

In order to reconstruct phylogenetic relationships among crossed species, we further used the GATK4 FastaAlternateReferenceMaker option to create locus-specific consensuses from each sample. Regions with low read depth in each new consensus were identified using the samtools software (*Danecek et al., 2021*) with *depth* option and the output file was rewritten into bed format where sites with depth <10 were masked by 'N' using bedtools (*Quinlan and Hall, 2010*) and *maskfasta* option. To mitigate the locus dropout in distant species, we than used custom R scripts (R Core Team 2020) with functionalities of the seqinr package *Charif and Lobry, 2007* to select the alignments where reads from all investigated species are present. Final phylogenetic analysis was thus based on 1960 loci with length >750 bp & > 30 parsimony informative sites, where all species had correctly read sequence variants on more than 70% of sites. Individual Maximum Likelihood gene trees were reconstructed by IQ-TREE v. 2.0.3 *Nguyen et al., 2015* using the extended model selection with free rate of heterogeneity in combination with 1000 ultrafast bootstrap replicates (*Kalyaanamoorthy et al., 2017*; *Hoang et al., 2018*). Consensus species tree was estimated by ASTRAL v. 5.5.6 (*Zhang et al., 2018*). Gene Concordance Factors (gCF) and site Concordance Factors (sCF) were estimated (*Minh et al., 2020*) and resulting trees were processed using the ape package (*Paradis and Schliep, 2019*) and plotted by DensiTree v2.0 (*Bouckaert, 2010*).

## Crossing experiments

Artificial spawning was performed by hormonal stimulation of fish. Mature fish (males and females) were injected twice in the peritoneal cavity (24 and 12 hr before fertilization) with hormone Ovopel (Interfish Kf) (*Tichopád et al., 2022*). First injection was applied with a solution of one Ovopel pill per 20 mL of 0.9% NaCl. Second injection was done with a solution of 1 pill per 5 mL of 0.9% NaCl. The ratio of injected solution in both cases was 0.05 mL per 10 g of fish weight. Fish eggs were gently squeezed 24 hr after first injection and transferred to a Petri dish (ø=8.6 cm, 1.5 cm high) with a density not higher than 100 eggs per dish. Male sperm was also obtained by squeezing or by homogenization of male testes from euthanized fish and transferred in Hanks' balanced salt solution (Sigma-Aldrich) or directly applied on loach eggs together with fresh water to activate spermatozoa. Incubation of eggs was performed at room temperature (23–25 C). After hatching, free larvae were transferred into plastic pots with the following size: 25×25 × 15 cm. Larvae were fed twice daily with nauplii of the brine shrimp *Artemia*. After 45 days from hatching, juveniles were fed with *Tubifex* worms once per day. Adult fish were fed with tablets of dry food when live food was not available. Starting from two weeks we randomly selected fish larvae for the whole mount investigation of fish gonads. For immunofluorescent staining we randomly selected juveniles with an age of two months after hatching. Analysis of diplotene chromosomes was performed on adult and subadult females older than 6 months. Adult and subadult males older than half a year were used for flow cytometry measurements of the cell suspension from their gonads, histology and pachytene chromosome analysis.

## DNA flow cytometry

DNA content of the cell suspension from testes was measured by BD FACSAria flow cytometer. Dissected testes were collected into Versene solution (Thermo Fisher Scientific). In total, we investigated 22 hybrid males from 19 families and two sexual species (EE). To release and stain the cell nuclei, testes were homogenized and incubated with 0.1% Triton X100, 10 μ/ml DAPI and 15 mM MgCl2 at +4 °C overnight. At least 10,000 events were analysed. The samples from testes of sexual diploid species EE, were used as an internal control for measurements. Data was further analysed by BD FACSDiva software (version 6.1.3). We assessed haploid cells to 1n1C peak, and diploid cells to 2n2C and 2n4C peak (2n2C cell population represent cells before DNA synthesis, while the 2n4C cell population represents cells after the S phase, possibly indicating meiocytes arrested during pachytene due to chromosomal mispairing in meiosis), where n represents a set of chromosomes, and C represents the number of chromatids.

## Preparation of mitotic chromosomes

Cell suspensions with mitotic chromosomes from all specimens were obtained either from kidneys and/or regenerated caudal fins according to *Majtánová et al., 2016*. Metaphase chromosomes were stained with Giemsa to check the morphology of chromosomes. Chromosome formula = 2 n (48-50) metacentric, submetacentric/acro- and subtelocentric.

## Pachytene chromosomes preparation and immunofluorescent staining

To obtain pachytene chromosomes we used protocols described in *Dedukh et al., 2020*; *Dedukh et al., 2021* for males and females, respectively. Gonads were manually homogenized in 1×PBS. In the case of males, we incubated 1 μl of suspension in 30 μl of hypotonic solution (1/3 of 1×PBS), preliminarily dropped on SuperFrost slides (Menzel Gläser) for 20 min. Subsequently slides were fixed in 2% paraformaldehyde (PFA) for 4 min. In the case of females, 20 μl of cell suspension was put on SuperFrost slides (Menzel Gläser) followed by addition of 40 μl of 0.2 M Sucrose and 40 μl of 0.2% Triron X100 for 7 min and subsequently fixed in 2% PFA for 16 min. After fixation, slides with male and female pachytene spreads were air dried, washed in 1×PBS and used for immunofluorescent staining.

During immunofluorescent staining we visualized synaptonemal complexes (SC) of pachytene chromosomes using rabbit polyclonal antibodies (ab15093, Abcam) against SYCP3 protein (the lateral component of SC) and chicken polyclonal antibodies (a gift from Sean M. Burgess) against SYCP1 (the central component of SC). Crossing-over foci were identified using mouse monoclonal antibodies (ab14206, Abcam) against MLH1 protein, a mismatch repair protein. Slides were incubating with 1% blocking reagent (Roche) in 1×PBS and 0.01% Tween-20 for 20 min. Primary antibodies (dilutions as recommended by manufacturers) were added for 1 hr at RT followed by three times washing in

1×PBS. Afterwards, corresponding secondary antibodies were added for 1 hr at RT: Alexa-594-conjugated goat anti-rabbit IgG (H+L) (Invitrogen), Alexa-594-conjugated goat anti-chicken IgG (H+L) (Invitrogen) and Alexa-488-conjugated goat anti mouse IgG (H+L) (Invitrogen). Slides were washed in 1×PBS with 0.05% Tween-20 and mounted in Vectashield/DAPI (1.5 mg/ml) (Vector, Burlingame, Calif., USA).

## Diplotene chromosome preparation

Diplotene chromosomes (so called lampbrush chromosomes) were prepared from parental and hybrid females accordingly *Dedukh et al., 2020*; *Dedukh et al., 2021*. Ovaries with vitellogenic oocytes of size 0.5–1.5 mm in diameter were taken from females and placed in OR2 saline (82.5 mM NaCl, 2.5 mM KCl, 1 mM MgCl$_2$, 1 mM CaCl$_2$, 1mM Na$_2$HPO$_4$, 5 mM HEPES (4-(2-hydroxyethyl)–1-piperazineethane sulfonic acid); pH 7.4). Individual oocytes were transferred to the isolation medium '5:1' (83 mM KCl, 17 mM NaCl, 6.5 mM Na$_2$HPO$_4$, 3.5 mM KH$_2$PO$_4$, 1 mM MgCl$_2$, 1 mM DTT (dithiothreitol); pH 7.0–7.2) for microsurgical isolation of nuclei. Nuclear envelopes were manually removed in one-fourth strength '5:1' medium with the addition of 0.1% PFA and 0.01% 1 M MgCl$_2$ in glass chambers attached to a slide. This procedure ensures the presence of a chromosomal set from an individual oocyte in each separate chamber. Slides with oocyte nuclei contents were centrifuged for 20 min at +4 °C, 4000 rpm, fixed for 30 min in 2% PFA in 1×PBS, and post-fixed in 50% and 70% ethanol. After drying chromosomal spreads were mounted in Vectashield/DAPI (1.5 mg/ml; Vector, Burlingame, Calif., USA).

## Whole-mount immunofluorescence staining

Gonads of fish larvae were isolated and fixed in 2% PFA in 1×PBS followed by washing in 1×PBS. Tissues were stored until usage in 1×PBS with the addition of 0.02% sodium azide. Prior to immunofluorescent staining, gonadal fragments were permeabilized in a 0.5% solution of Triton X100 in 1×PBS for 4–5 hr at RT and washed in 1×PBS at RT. The following primary antibodies were used: rabbit polyclonal antibodies DDX4 antibody (C1C3, GeneTex) against vasa protein; rabbit polyclonal (ab14206, Abcam) against SYCP3 protein and chicken polyclonal against SYCP1 protein (a gift from Sean M. Burgess). After incubation for 1–2 hr in a 1% blocking solution (Roche) dissolved in 1×PBS, primary antibodies were added for 12 hr at RT. After three times washing in n 1×PBS with 0.01% Tween (ICN Biomedical Inc), secondary antibodies Alexa-488-onjugated goat anti-rabbit IgG (H+L) (Invitrogen) and Alexa-594-conjugated goat anti-chicken IgG (H+L) (Invitrogen) were added for 12 hr at RT. After three times washing in 1×PBS with 0.01% Tween (ICN Biomedical Inc) tissues were stained with DAPI (1 mg/ml) (Sigma) in 1×PBS at RT overnight.

## Confocal laser scanning microscopy

Tissues fragments were placed in a drop of Vectashield antifade solution containing 1 mg/ml DAPI. Confocal laser scanning microscopy was carried out using a Leica TCS SP5 microscope based on the inverted microscope Leica DMI 6000 CS (Leica Microsystems, Germany). Specimens were analyzed using HC PL APO 40 x objective. Diode and helium-neon lasers were used to excite the fluorescent dyes DAPI and Cy3, respectively. The images were captured and processed using LAS AF software (Leica Microsystems, Germany).

## Wide-field and fluorescence microscopy and image processing

Mitotic chromosomes were examined by Olympus BX53 epifluorescence microscope and Axio Imager Z2 microscope equipped with CCD camera (DP30W Olympus) and CoolCube 1 b/w digital camera (MetaSystems, Altlussheim, Germany), respectively. Axio Imager Z2 epifluorescence microscope is equipped by Metasystems platform for automatic search, capture and image processing. Meiotic chromosomes were analysed using Olympus BX63 microscopes equipped with standard fluorescence filter sets, and sCMOS camera (Prime95B Teledyne Photometrics) using CellSense Dimension software (Olympus). The IKAROS imaging program (Metasystems, Altlussheim, Germany) were used to analyse grey-scale images. The captured digital images from FISH experiments were pseudocoloured (blue for DAPI, red for Alexa-594, green for Alexa-488) and superimposed using Microimage and Adobe Photoshop software, version CS5, respectively. Microphotographs were finally adjusted and arranged in Adobe Photoshop CS6 software.

## Karyotype differences between species

Chromosomal data for each species was obtained from the literature (*Cataudella et al., 1977*; *Hnátková et al., 2018*; *Janko et al., 2005*; *Kim and Lee, 1990*; *Kim et al., 1995*; *Marta et al., 2020*; *Suzuki and Taki, 1982*; *Ueno and Ojima, 1976*) and present work (*Figure 1—figure supplement 2*, *Supplementary file 1*). The inspection of karyotypes was done visually and we noticed the numbers of meta-/submeta- and subtelo-/acrocentrics for each species. The magnitude of the autosomal karyotypic differences between focal species was calculated using the autosomal karyotype index (AKD) (*Castiglia, 2014*), which is calculated as the sum of absolute differences in diploid numbers of chromosomes (2 n) divided by two and the absolute differences in the autosomal fundamental numbers of arms (NF) also divided by two. The data are presented in *Supplementary file 2*.

## Statistical analyses

To investigate whether karyotype divergence among species correlate with their genetic divergence, we used a Mantel test with 10,000 replicates to compare a matrix of pairwise chromosomal divergences expressed in pairwise AKD index and a matrix of genetic divergences expressed in pairwise exome-wide SNP distances.

We than investigated whether proportions of endoreplicated oocytes at the pachytenic stage differ among various types of F1 crosses. To do so, we compared the counts of duplicated and non-duplicated oocytes in every female using the generalized linear mixed effect model (GLMM) with individuals taken as a random factor and binomial error distribution. Model significance was evaluated with the Wald Z-test implemented in the R library CAR (*Fox and Friendly, 2007*).

Finally, we tested whether numbers of bivalents formed by F1 hybrids are explicable by the amount of karyotype differences measured by the AKD index of morphological differentiation among their parental species. To do so, we fitted the numbers of bivalents to hybrid's sex and AKD as well as SNP distances between parental species by a GLMM model with individuals taken as a random factor and Poisson distribution of error in R library LME4 (*Bates et al., 2009*). We performed the forward selection with each pair of nested model tested by analysis of deviance in GLMs, including their interaction with hybrids' sex. Z values and Pr(>|z|) are reported as z and p, respectively.

## Acknowledgements

The authors thank Šárka Pelikánová (Institute of Animal Physiology and Genetics), Daniel Kulik and Grzegorz Skórzewski (Museum of Natural History, University of Wroclaw) for helping with taking care of the fish; Ruxanda Cucerenco (Nicolae Testemițanu State University of Medicine and Pharmacy, Chisinau) and Anton Svinin (University of Tyumen) for helping with preparation of samples with immunofluorescent staining, Sean Burgess (College of Biological Sciences, University of California) for providing primary antibodies against SYCP1; Vladislav Vasiullin for the help with the preparation of illustrations; Stephen A Schlebusch (Charles University, Prague) and Roger Butlin (University of Sheffield, Sheffield) for reading the first draft of the manuscript. This study was supported by following grants: Czech Science Foundation grant Nos. 19–21552 S, 21–25185 S, RVO 67985904, and Ministry of Education, Youth and Sports of the Czech Republic grant No. 539 EXCELLENCE CZ.02.1.01/0.0/0.0/15_0 03/0000460 OP RDE. (AM, DD, KJ). Ministry of Education and Research in Moldova grant DIVZOO 20.80009.7007.12 (AM).

## Additional information

### Funding

| Funder | Grant reference number | Author |
| --- | --- | --- |
| Grantová Agentura České Republiky | 19-21552S | Karel Janko |
| Grantová Agentura České Republiky | 21-25185S | Karel Janko |

| Funder | Grant reference number | Author |
| --- | --- | --- |
| Grantová Agentura České Republiky | RVO 67985904 | Karel Janko |
| Ministerstvo Školství, Mládeže a Tělovýchovy | grant No. 539 EXCELLENCE CZ.02.1.01 /0.0/0.0/15_003/0000460 OP RDE | Anatolie Marta |
| Ministerul Educației, Culturii și Cercetării | DIVZOO 20.80009.7007.12 | Anatolie Marta |

The funders had no role in study design, data collection and interpretation, or the decision to submit the work for publication.

## Author contributions

Anatolie Marta, Conceptualization, Formal analysis, Investigation, Visualization, Methodology, Writing – original draft, Writing – review and editing; Tomáš Tichopád, Software, Formal analysis; Oldřich Bartoš, Data curation, Software, Formal analysis; Jiří Klíma, Mujahid Ali Shah, Karel Halačka, Lukáš Choleva, Methodology; Vendula Šlechtová Bohlen, Formal analysis, Methodology; Joerg Bohlen, Conceptualization, Writing – review and editing; Matthias Stöck, Writing – review and editing; Dmitrij Dedukh, Data curation, Formal analysis, Investigation, Visualization, Methodology, Writing – original draft, Writing – review and editing; Karel Janko, Data curation, Formal analysis, Funding acquisition, Project administration, Writing – review and editing, Supervision

## Author ORCIDs

Anatolie Marta ⓘ http://orcid.org/0000-0002-4457-8838
Tomáš Tichopád ⓘ http://orcid.org/0000-0002-9154-2969
Oldřich Bartoš ⓘ http://orcid.org/0000-0001-5441-5592
Mujahid Ali Shah ⓘ https://orcid.org/0000-0001-5727-7164
Matthias Stöck ⓘ http://orcid.org/0000-0003-4888-8371
Dmitrij Dedukh ⓘ http://orcid.org/0000-0002-1152-813X
Karel Janko ⓘ http://orcid.org/0000-0002-7866-4937

## Ethics

The Valid Animal Use Protocol was in force during the study at the Institute of Animal Physiology and Genetics, Libechov, Czech Republic (No. CZ 02386). All institutional and national guidelines were covered by the "Valid Animal Use Protocol" No. CZ 02386 of the Laboratory of Fish genetics.

Reviewer #1 (Public Review): https://doi.org/10.7554/eLife.88366.3.sa1
Reviewer #2 (Public Review): https://doi.org/10.7554/eLife.88366.3.sa2
Author Response https://doi.org/10.7554/eLife.88366.3.sa3

# Additional files

## Supplementary files

• Supplementary file 1. This study uses a summary of karyotypes and RAG1 sequence of loaches (Cobitidae). The references are indicated for species karyotypes that were previously described. The Karyotype of *C. ohridana* is published for the first time in this work.

• Supplementary file 2. The numbers of analyzed male and female *Cobitis* samples from each F1 obtained family. Most specimens were used for the pachytene analysis in order to increase the chance to observe duplicated cells in females, as their incidence per slide is very low in clonal hybrids. Also, we used as many individuals as possible for this approach to have a statistically significant number of SC per individual.

• Supplementary file 3. Dataset with every SC reported from all analyzed specimens. All this information was used in order to generate *Figure 5*.

• MDAR checklist

## Data availability

The genomic sequence data generated and analyzed during the current study are available in the NCBI repositories with the bioproject ID PRJNA1025348 and the following accession numbers: C. elongatoides: SAMN37721543, C. taenia: SAMN37721544, C. taurica: SAMN37721545, C. tanaitica: SAMN37721546, C. pontica: SAMN37721547, C. ohridana: SAMN37721548, C. strumicae: SAMN37721551, C. bilineata: SAMN37721552.

The following datasets were generated:

| Author(s) | Year | Dataset title | Dataset URL | Database and Identifier |
|---|---|---|---|---|
| Janko K | 2023 | Model organism or animal sample from Cobitis elongatoides | https://www.ncbi.nlm.nih.gov/biosample/SAMN37721543 | NCBI BioSample, SAMN37721543 |
| Janko K | 2023 | Model organism or animal sample from Cobitis taenia | https://www.ncbi.nlm.nih.gov/biosample/SAMN37721544 | NCBI BioSample, SAMN37721544 |
| Janko K | 2023 | Model organism or animal sample from Cobitis taurica | https://www.ncbi.nlm.nih.gov/biosample/SAMN37721545 | NCBI BioSample, SAMN37721545 |
| Janko K | 2023 | Model organism or animal sample from Cobitis tanaitica | https://www.ncbi.nlm.nih.gov/biosample/SAMN37721546 | NCBI BioSample, SAMN37721546 |
| Janko K | 2023 | Model organism or animal sample from Cobitis pontica | https://www.ncbi.nlm.nih.gov/biosample/SAMN37721547 | NCBI BioSample, SAMN37721547 |
| Janko K | 2023 | Model organism or animal sample from Cobitis ohridana | https://www.ncbi.nlm.nih.gov/biosample/SAMN37721548 | NCBI BioSample, SAMN37721548 |
| Janko K | 2023 | Model organism or animal sample from Cobitis strumicae | https://www.ncbi.nlm.nih.gov/biosample/SAMN37721551 | NCBI BioSample, SAMN37721551 |
| Janko K | 2023 | Model organism or animal sample from Cobitis bilineata | https://www.ncbi.nlm.nih.gov/biosample/SAMN37721552 | NCBI BioSample, SAMN37721552 |

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
