## [Editor Report · eLife assessment]

This paper provides **important** insights into how asexual reproduction can arise in interspecific hybrids. The evidence supporting the conclusions is **compelling**, with rigorous molecular cytogenetic experiments showing the production of clonal gametes is common across hybrids between closely to moderately divergent sexual species. By highlighting the potential for asexuality to evolve in hybrids during a relatively wide window of species divergence, this work will be of broad interest to evolutionary biologists.

---

## [Referee Report · Reviewer #1 (Public Review)]

This paper provides new evidence on the relationship between genetic/chromosome divergence and capacity for asexual reproduction (via unreduced, clonal gametes) in hybrid males or females. Whereas previous studies have focussed just on the hybrid combinations that have yielded asexual lineages in nature, the authors take an experimental approach, analysing meiotic processes in F1 hybrids for combinations of species spanning different levels of divergence, whether or not they form asexual lineages in nature. As such, the findings here are a substantial advance towards understanding how new asexual lineages form.

The quality of the work is high, the analyses are sound, and the authors sensibly link their observations to the speciation continuum. I should also add that the cytogenetic work here is just beautiful!

A key finding is that the precondition for asexual reproduction - the formation of unreduced gametes - is not unusual among hybrid females, so that we have to consider other factors to explain the rarity of asexual species - a major unresolved issue in evolutionary biology. This work also highlights a previously overlooked effect of chromosome organisation on speciation.

---

## [Referee Report · Reviewer #2 (Public Review)]

The authors investigate the origin of asexual reproduction through hybridization between species. In loaches, diploid, polyploid, and asexual forms have been described in natural populations. The authors experimentally cross multiple species of loaches and conduct an impressively detailed characterization of gametogenesis using molecular cytogenetics to show that although meiosis arrests early in male hybrids, a subset of cells in females undergo endoreplication before meiosis, producing diploid eggs. This only occurred in hybrids of parental species that were of intermediate divergence. This work supports an expanding view of speciation where asexuality could emerge during a narrow evolutionary window where genomic divergence between species is not too high to cause hybrid inviability, but high enough to disrupt normal meiotic processes.

I enjoyed reading this study and I was impressed by the rigorous experiments. The authors provide strong evidence that premeiotic genome endoreplication is the mechanism behind asexually-reproducing females. In addition, I found the evidence convincing that this phenomenon is a consequence of combining two divergent genomes in an F1 hybrid female. The authors did not observe a single incidence of genome duplication in any of the parental species among a large number of surveyed oocytes.

---

## [Author Response]

The following is the authors’ response to the original reviews.

Thank you very much for the kind comments about our manuscript. We have improved the text to address all reviewers’ comments and suggestions. Additionally, we corrected and improved the supplementary tables.

**Reviewer #1 (Public Review):**
This paper provides new evidence on the relationship between genetic/chromosome divergence and capacity for asexual reproduction (via unreduced, clonal gametes) in hybrid males or females. Whereas previous studies have focussed just on the hybrid combinations that have yielded asexual lineages in nature, the authors take an experimental approach, analysing meiotic processes in F1 hybrids for combinations of species spanning different levels of divergence, whether or not they form asexual lineages in nature. As such, the findings here are a substantial advance towards understanding how new asexual lineages form.The quality of the work is high, the analyses are sound, and the authors sensibly link their observations to the speciation continuum. I should also add that the cytogenetic work here is just beautiful!A key finding is that the precondition for asexual reproduction - the formation of unreduced gametes - is not unusual among hybrid females, so that we have to consider other factors to explain the rarity of asexual species - a major unresolved issue in evolutionary biology. This work also highlights a previously overlooked effect of chromosome organisation on speciation.

Thank you for the nice comments about our work as well as for appreciating our cytogenetics work and figures.

**Reviewer #2 (Public Review):**
The authors investigate the origin of asexual reproduction through hybridization between species. In loaches, diploid, polyploid, and asexual forms have been described in natural populations. The authors experimentally cross multiple species of loaches and conduct an impressively detailed characterization of gametogenesis using molecular cytogenetics to show that although meiosis arrests early in male hybrids, a subset of cells in females undergo endoreplication before meiosis, producing diploid eggs. This only occurred in hybrids of parental species that were of intermediate divergence. This work supports an expanding view of speciation where asexuality could emerge during a narrow evolutionary window where genomic divergence between species is not too high to cause hybrid inviability, but high enough to disrupt normal meiotic processes.

Thank you.

I enjoyed reading this study and I appreciate the amount of work it takes to conduct these types of cytogenetic experiments. But, my main concern with this study is I was left wondering if the sample sizes are large enough to get a sense how variable endoreplication is in these loach species. Most of the hybrids between species are the result of crosses between 1-2 families. Within males and females, meiocyte observations are limited to a handful of pachytene and diplotene stages. I think it would be helpful to be more transparent about the sample sizes in the main text.

Thank you for raising this point. We have improved the Supplementary Tables S2 and S3 to clarify how many individuals we analyzed from each genetic family and added this information to the main text. In total we obtained 12 combinations with 19 F1 hybrid families. For the combination, C. elongatoides x C. taenia hybrids we obtained three families, for C. elongatoides x C. ohridana, C. elongatoides x C. tanaitica, C. elongatoides x C. bilineata and C. ohridana x C. bilineata, we obtained two families For the rest of the combinations of hybrids we obtained single family. From these families, 79 individuals were used for the analysis of the meiocites. Additionally, 24 parental individuals, males and females, were analysed. For the parental species, we analysed 852 cells, for hybrid males we investigated 244 cells, and 665 cells for hybrid females.

Along these lines, the authors argue against the possibility that endoreplication may be predisposed to occur at a higher rate in some species (line 291). Instead, they suggest that endoreplication is a result of perturbing the cell cycle by combining the genomes of two different species. Their main argument is based on gonocyte counts from parental females in a previous reference. It is essential to include counts from the parents used in this study to make a clear comparison with the F1s.

Thank you, we agree with your comment and included the observations of meiocytes from several parental species, i.e. C. elongatoides, C. taenia, C. pontica, C. tanaitica, and C. ohridana. Among 852 cells analyzed, we did not observe cells with duplicated genomes and abnormalities in chromosomal pairing. By contrast, among 665 pachytene cells of F1 hybrid females, we revealed altogether ~1% of endoreplicated ones. We tested these data by binomial GLM and found these differences to be significant, suggesting that sexuals, even if they may have some unnoticed duplication events, clearly have a significantly lower incidence of abnormal pachytene cells. We have now included this information in the main text.

In the discussion (lines 320-333), the authors postulate the sex-specific clonality they observe could be a result of Haldane's rule. Given these fish do not have known sex chromosomes, I do not find this argument strong. Haldane's rule refers to the exposure of recessive incompatibilities with the sex chromosomes in the hybrid heterogametic sex. This effect would therefore be limited to degenerated sex chromosomes where much of the sequence content on the Y or W has been lost. These species may have homomorphic sex chromosomes, but if this is the case, they likely are not very degenerated. Instead, it seems more plausible that the sex-specific effect the authors observe is due to intrinsic differences of spermatogenesis and oogenesis. Is there any information about sex-specific differences in the fidelity of gametogenesis from other species that would support a higher likelihood of endoreplication?

Thank you for this important question, however, we think it was a misunderstanding. We do not postulate that our observation conforms to Haldanes’ rule as, by contrast to this rule based on sex chromosomes, our previous publication demonstrated that whatever the gonadal sex differentiation is in our taxa, the ability to overcome sterility by asexual gametogenesis is always confined to female gonadal environment (or oogenesis in general), even in the transplanted spermatogonial cells (Tichopad et al. 2022). What we meant by our text is that our results do not fully conform to Haldane’s rule. We therefore reworded our text to rule out such a misconception.

Nonetheless, we note that it has been demonstrated that Haldanes’ rule is also applicable to species with little differentiated sex chromosomes (e.g. Presgraves and Orr 1998) and that recessive incompatibilities are not the only explanation as faster male theory or faster X may also apply in such cases (Dufresnes et al. 2016). Therefore, we have kept our remarks about Haldane’s rule here. Moreover, for several parental species, we preliminary found the occurrence of an XY gonadal sex differentiation system, albeit these are unpublished and need further validation.

The final thing I was left wondering about was this missing link between endoreplication and activating the embryonic development of the diploid egg. In these loach species, a sperm is required to activate egg development, but the sperm genome is discarded (line 100). What is the mechanism of this and how does it evolve concurrently during hybridization?

Thank you for the comment. There have been many speculations about why gynogens actually need sperm to activate their egg development, but to our knowledge, no explanation has been validated to date. Interestingly, a recent theoretical model by Fyon et al. BiorXiv 2023 suggested that the ability of sperm exclusion may evolve separately from the ability to produce clonal eggs. Hence, this topic is complex and remains unresolved, and we feel that it is out of the scope of the present MS. We have slightly modified the text and added 2 refs., to address your suggestion.

**Reviewer #1 (Recommendations For The Authors):**
The paper is well prepared - though the resolution of Fig 1 on the pdf is rather poor.

Thank you! We have now provided the high-resolution figures.

Overall, I have few suggestions for improvements:Line 58. How does endoduplication itself "overcome accumulated incompatibilities" other than failure of synapsis? Perhaps by maintaining the F1 state, and so avoiding reduced fitness arising from recombination and disruption of coadapted gene combinations.

We have added a sentence to the main text “Premeiotic genome endoreplication thus not only ensures clonal reproduction but also allows hybrids to overcome problems in chromosome pairing that would otherwise lead to their sterility 15,17.” that we hope sufficiently addresses this issue.

Line 118 - please explain the AKD index here - as you have some in SI. Also please be clearer on how you measure genetic divergence as proportion of heterozygous SNPs - presumably this is via exon sequences from F1 females?

Please note that we have explained the AKD index in the relevant part of the Methods section already. However, we have now also added a brief explanation to the Results section, as suggested. We apologize for imprecise description of the genetic divergence measurements. As described in the Methods section, this is not measured by heterozygosity (as we wrongly stated here), but as p-distance among sequences of coding regions between parental species.

Lines 126 ff. It is unfortunate that the design of the crosses was not more balanced or extensive. Nonetheless, I do appreciate the effort involved here and think the results are solid as is.

Thank you.

Line 142. Please define PS and TB (and other acronyms) at first use.

We have added the definition for all acronyms at the first use.

Lines 192-193. What about EP and EN - as shown to have unreduced gametes in Fig. 2?

Thank you for this question. Based on analyses of the diplotene stage, we showed that EP and EN hybrids produced diploid eggs. However, in pachytene, we did not find duplicated oocytes due to the rarity of endoreplication. Similarly, the low incidence of duplicated pachytene cells was observed in natural as well as F1-hybrids in loaches and reptiles (Newton et al., 2016, Dedukh et al., 2021, 2022).

Lines 217-219. The observed correlation of chromosome divergence (AKD index) and numbers of bivalents in pachytene makes sense and is an important observation. Did this GLM simultaneously consider the effect of genetic divergence (as implied in methods)?

Thank you for this comment. We originally tested separately the fit of two models, one with AKD and the other with SNP divergence. Since the AKD model significantly outperformed the SNP-based one, we focused our interpretation on the former. However, as you suggested, we now re-calculated the model taking into account the joint effects of both predictors in a single model and indeed, this model outperformed both single predictors. In conclusion, while AKD is still the strongest single predictor for the observed amounts of bivalents, the additional effect of genetic distance still significantly improves the model fit. We have now included this result into the main text.

This finding does not alter our conclusions, it just suggests that the effect of chromosomal morphology is probably more complex, involving the role of more subtle sequence divergence or structural variants.

Line 242. The Discussion is a great read - careful interpretation and a really interesting interpretation in context of the broader literature.

Thank you for the appreciation. Your positive feedback and evaluation are highly motivating us to expand our work.

Line 396. Some references from book chapters (18, 52) are incomplete. Please fix.

We have now corrected these references accordingly.

**Reviewer #2 (Recommendations For The Authors):**
Transparency about meiocyte sample sizes: These counts are all in supplemental table 3. From this table, it is unclear if a majority of these meiocytes are from a single individual or from multiple males or females. Or, in the crosses where there are multiple families, are the meiocytes sampled from all families? I am trying to get a sense whether endoreplication and the fidelity of oogenesis could be influenced by genetic variants segregating within species. If the meiotcytes are only sampled from a single individual from a single cross, you may not see this variation. If this is the case, perhaps the correlation between genetic divergence and the formation of asexual clones may not be as strong. Additional replicates may not be feasible, but at a minimum I think it would be helpful to address whether endoreplication could or could not be variable and if the sample sizes are sufficient.

Thank you for raising this point. We have improved the Supplementary table to clarify how many individuals we analyzed from each family and added this information to the main text. Unfortunately, additional replicates are not feasible due to the long generation time of the fish. We otherwise agree with your comment and included this point in the Discussion.

Gonocyte counts from parental females: The authors say they "analysed hundreds of gonocytes of sexual females without a single incidence of genome endoreplication." I could not find a clear count in the references given. They note that the incidence of endoreplication was very low in pachytene cells in this study (0.7%).

Thank you, we agree with your comment and included the observations of meiocytes from several parental species, i.e. C. elongatoides, C. taenia, C. pontica, C. tanaitica, and C. ohridana. Among 852 cells analyzed, we did not observe cells with duplicated genomes and abnormalities in chromosomal pairing. By contrast, among 665 pachytenic cells of F1 hybrid females, we revealed altogether ~1% of endoreplicated ones. We tested these data by binomial GLM and found these differences to be significant, suggesting that sexuals, even if they may have some unnoticed duplication events, clearly have significantly lower incidence. of abnormal pachytene cells. We have now included this information in the main text.

They refer to supplemental table 4 (line 196), which does not exist in the supplement. The authors should report these numbers in the revised manuscript.

Thank you for pointing this out. We have corrected the name of the supplementary table, it actually is supplementary table S3.